# Immunomodulatory and Antioxidant Properties of a Novel Potential Probiotic *Bacillus clausii* CSI08

**DOI:** 10.3390/microorganisms11020240

**Published:** 2023-01-18

**Authors:** Ekaterina Khokhlova, Joan Colom, Annie Simon, Shahneela Mazhar, Guillermo García-Lainez, Silvia Llopis, Nuria Gonzalez, María Enrique-López, Beatriz Álvarez, Patricia Martorell, Marta Tortajada, John Deaton, Kieran Rea

**Affiliations:** 1Deerland Ireland R&D, Ltd., ADM, Bio-Innovation Unit, Rm. 331 Food Science Building, College Rd., University College Cork, T12 K8AF Cork, Ireland; 2Archer Daniels Midland, Nutrition, Health&Wellness, Biopolis S.L. Parc Científic Universitat de València, C/ Catedrático Agustín Escardino Benlloch, 9, 46980 Paterna, Spain; 3Deerland Probiotics & Enzymes, ADM, Science and Technology Department, 3800 Cobb International Blvd., Kennesaw, GA 30152, USA

**Keywords:** *Bacillus clausii*, spore-based probiotics, immunomodulation, adherence, antioxidant

## Abstract

Spore-forming bacteria of the *Bacillus* genus have demonstrated potential as probiotics for human use. *Bacillus clausii* have been recognized as efficacious and safe agents for preventing and treating diarrhea in children and adults, with pronounced immunomodulatory properties during several in vitro and clinical studies. Herein, we characterize the novel strain of *B. clausii* CSI08 (Munispore^®^) for probiotic attributes including resistance to gastric acid and bile salts, the ability to suppress the growth of human pathogens, the capacity to assimilate wide range of carbohydrates and to produce potentially beneficial enzymes. Both spores and vegetative cells of this strain were able to adhere to a mucous-producing intestinal cell line and to attenuate the LPS- and Poly I:C-triggered pro-inflammatory cytokine gene expression in HT-29 intestinal cell line. Vegetative cells of *B. clausii* CSI08 were also able to elicit a robust immune response in U937-derived macrophages. Furthermore, *B. clausii* CSI08 demonstrated cytoprotective effects in in vitro cell culture and in vivo *C. elegans* models of oxidative stress. Taken together, these beneficial properties provide strong evidence for *B. clausii* CSI08 as a promising potential probiotic.

## 1. Introduction

The term probiotics acknowledged by the Agriculture Organization of the United Nations and the WHO (FAO/WHO) in 2001 and revisited later, in 2014, defines them as “live microorganisms that, when administered in adequate amounts, confer a health benefit on the host” [1]. The growing body of evidence highlighting the fundamental role of microbiota in human health and disease [2] drives global interest in probiotic research, and novel approaches helping to maintain “healthier” gut microbiota are in demand. Indeed, health benefits of probiotics have been shown in multiple clinical trials and are further backed by meta-analyses and systematic reviews. Through various modalities, including interacting with the host microbiota and the host intestinal epithelium, probiotics can influence a number of physiological processes including metabolism, immunity and endocrine function amongst others [3].

Probiotic strains most cited in biomedical literature belong to the genera bifidobacteria and lactobacilli [1,4]. In this context, spore-forming bacteria of the genus *Bacillus* are clearly understudied [5], yet a number of bacilli strains have been shown to possess probiotic characteristics and have decades-long history of medical use [5,6]. The ability to form endospores that are resistant to heat, UV radiation, desiccation [7], can tolerate low pH and high concentrations of bile salts [8] confers Bacillus-based probiotics several unique features. Being administered in the spore state, *Bacillus* spp. can survive aggressive gastric conditions and reach the small intestine in large numbers where they germinate into vegetative form [9]. Spore-based probiotic formulations have a long shelf life and do not require refrigeration during storage [10,11]. Moreover, *Bacillus* spp. spores showed decent stability during processing and storage of functional foods, such as pastries, fruit juices and preserves, sausages, and pasta [12,13,14].

Different strains belonging to nine *Bacillus* species such as *B. subtilis*, *B. coagulans*, *B. indicus*, *B. licheniformis, B. clausii* have been recognized as probiotics for human use [15]. *B. clausii* strains are ubiquitous in nature, and have been isolated from soil [16], seawater [17], milk [18] amongst others. Importantly, isolates of *B. clausii* have been recovered from human feces more frequently than other *Bacillus* species [19].

Safety and efficacy of different probiotic strains of *B. clausii* have been demonstrated in a number of clinical trials conducted around the globe [20,21,22,23,24]. For example, *B. clausii* (O/C, N/R, SIN and T) can reduce the incidence of diarrhea, nausea, and epigastric pain associated with antibiotic treatment for *H. pilory* in adults [25,26]. Similarly, *B clausii* (*A. clausii* 088AE) showed their efficacy for managing diarrhea caused by broad-spectrum antibiotics in both children and adults [27]. Treatment with probiotic strains of *B. clausii* were also effective in reducing main clinical symptoms of acute diarrhea, including rotavirus-associated diarrhea in children [28,29,30,31]. In a recently published prospective study, *B. clausii* (O/C, N/R, SIN and T) was beneficial for managing symptoms of IBS in children [31]. Moreover, prophylactic administration of *B. clausii* to preterm infants was associated with faster achievement of full feeds [32]. In a pilot trial involving children with recurrent respiratory infections prolonged treatment with *B. clausii* has been proven to reduce the duration of infections [33]. *B. clausii* treatment has also been shown to alleviate symptoms of aphthous ulcer and oral candidiasis [34].

In vitro and preclinical studies of probiotic strains of *B. clausii,* as well as genome sequencing reports, revealed their key attributes, including the ability to produce antimicrobial substances and vitamins, the ability to withstand acidic pH and bile salts, and stress adaptation factors and putative adhesion proteins [35,36,37,38,39]. Interestingly, the immunomodulatory capacity of various *B. clausii* strains has been highlighted in a number of studies. Park et al. reported that supplementation with viable cells *B. clausii* KCTC 10277 BP can suppress pulmonary inflammation in ovalbumin-sensitized mice, reducing Th2 immune response and affecting hypoxia-related pathway in lung tissue [40]. *B. clausii* MTCC-8326 has been shown to protect murine RAW 264.7 macrophages against *S. typhimurium* infection supposedly by causing controlled inflammatory response and inducing several defensins and interferons [41]. Administration of *B. clausii* (O/C, N/R, SIN, and T) can affect expression of genes involved in inflammation, immune response, defense response etc. in duodenal mucosa of healthy individuals [42]. An in vitro study conducted on Swiss and C57 Bl/6j murine cells demonstrated the ability of the same strains to induce NOS II synthetase activity, IFN-γ production, and CD4+ T-cell proliferation [43].

MuniSpore^®^ is a commercial preparation of the spores *B. clausii* CSI08. The aim of the present work was to characterize this novel strain. Firstly, we evaluated general probiotic attributes of *B. clausii* CSI08, such as resistance to gastrointestinal conditions, adhesion capability, enzymatic profile, and antimicrobial activity. Additionally, we examined immunomodulatory and antioxidant properties of the vegetative form of the strain in in vitro and in vivo models.

## 2. Materials and Methods

### 2.1. Bacterial Strains and Sample Preparation

Munispore^®^, the single-strain spore-based probiotic *Bacillus clausii* CSI08, was provided by Deerland Probiotics and Enzymes (US). Vegetative form of *B. clausii* CSI08, *Escherichia coli* ATCC 25922, *Salmonella enteritidis* ATCC 13076, *Staphylococcus aureus* str. RF122, *Pseudomonas aeruginosa* DSM 3227 were cultured routinely at 37 °C in Trypticase soy broth (TSB) or agar (Merck, Ireland), unless otherwise specified. When preparing liquid cultures, above-named bacterial strains were grown with vigorous shaking (~170 rpm). Commercial probiotic strains *Lactobacillus rhamnosus* GG (ATCC 53103) and *Lactobacillus fermentum* were grown using DeMan-Rogosa-Sharpe broth or agar (Oxoid Ltd., Basingstoke, United Kingdom) at 37 °C.

For cell culture experiments, 10 mL of overnight cultures prepared under conditions described above were transferred into 50 mL falcon tubes and centrifuged for 10 min at 4000× *g*. Supernatants were then removed, and bacterial pellets were washed once with 10 mL of serum-free DMEM. Washed pellets were resuspended in 10 mL of serum and additives free high glucose DMEM. To obtain cell-free supernatants bacterial preparations in DMEM were further cultivated for 18 h. After that, cultures were centrifuged for 10 min at 4000× *g*; pellets were discarded, and supernatants were filter-sterilized using syringe-mounted filters with 0.22 µm pore diameter. 1N NaOH was added to cell-free supernatants to bring pH to ~7.5, if necessary.

### 2.2. Stability of B. clausii CSI08 Spores during Pasteurization

Spore suspension was prepared by adding 50 mg of Munispore^®^ spore powders to 100 mL of 1X PBS, pH 7.6 (Sigma-Aldrich, St. Louis, MO, USA). The suspension was mixed using a vortex for 2 min and dispensed into glass test tubes, 5 mL of suspension per each. The test tubes were treated at 45 °C, 75 °C and 90 °C in a water bath for 0.5, 1 or 3 min. Following incubation, samples were serially diluted and plated in 3M™ Petrifilm™ aerobic count plates.

### 2.3. Resistance of B. clausii CSI08 Spores to Simulated Gastric and Intestinal Conditions

The tolerance of *B. clausii* CSI08 spores to an in vitro simulated gastric and intestinal conditions were assayed following the publication by Pisano et al. [44] with some modifications. Briefly, 100 µL of spore suspensions in PBS or 100 µL of *L. rhamnosus* suspension in PBS were added to 900 µL of 0.3% pepsin (*w/v*, Sigma P6887) in NaCl pH 3 and incubated for 2 h at 37 °C in a water bath. Following incubation, spores and bacteria were pelleted by centrifuged at 3000× *g* for 5 min; the pellets were washed two times with 1 mL PBS and resuspended in 100 µL PBS. 900 µL of solution containing 0.1% pancreatin (*w/v*, Sigma P1750) and 0.3% bile salts (*w/v*, Sigma cat no. 48305) in PBS, pH 7.5, were added to the spores and bacteria and incubated for another 2 h at 37 °C in a water bath. After 1, 2, 3, and 4 h of incubation, samples’ aliquots were serially diluted and plated onto 3M™ Petrifilm™ aerobic count plates (*B. clausii* CSI08 spores) or MRS agar plates (*L. rhamnosus*).

### 2.4. Quantitative Analysis of Amino Acids and Vitamins in B. clausii CSI08 Supernatants

Overnight cultures of *B. clausii* CSI08 prepared in TSB were taken for analysis. Amino acids quantification was carried out using high performance liquid chromatography (HPLC) and fluorescence detection (FLD), with a precolumn derivatization step using 6-aminoquinolyl-*N*-hydroxysuccinimidyl carbamate (AQC). Regarding the determination of vitamins, ultra-high performance liquid chromatography (UHPLC) coupled to a simple quadrupole mass detector (MS) was used. Full technical description of the methods can be found in Appendix B and Appendix C.

### 2.5. Semi-Quantitative Assays for Carbohydrate Fermentation and Hydrolytic Activities

The ability of *B. clausii* CSI08 to use different carbon sources was investigated with API 50CH (BioMérieux, Hampshire, UK); hydrolytic activities were determined using the API-ZYM kit system (BioMérieux, Hampshire, UK), according to the instructions provided by manufacturer.

### 2.6. Antimicrobial Activity of B. causii CSI08 in Liquid Media

Tests in liquid culture were performed as previously described with some modifications [45]. Briefly, 10 µL of TSB were co-inoculated with 1 × 10^5^ CFU/mL of the test pathogen strain and 1 × 10^7^ CFU/mL of MuniSpore^®^. Co-cultures were incubated at 37 °C with vigorous shaking (170 rpm) for 24 h; after that the co-cultures were serially diluted and plated. Enumeration of gram-negative bacteria was performed on MacConkey agar (Merck, Ireland). *S. aureus* enumeration was performed on Mannitol egg yolk polymyxin agar (MYP, Merck, Ireland). Both MacConkey and MYP agar inhibited the growth of *B. clausii* CSI08, enabling the detection of each pathogen.

### 2.7. Determination of Antioxidant Activity In Vitro

Fifty milliliters of overnight cultures of *B. clausii* CSI08 and *L. rhamnosus* prepared in TSB broth and normalized by number of bacteria (1 × 10^8^ CFU/mL), were centrifuged at 4000× *g* for 15 min. Obtained pellets were washed three times with 10 mL phosphate buffered saline (PBS, Sigma-Aldrich) and resuspended in 1 mL PBS. Cell suspensions were transferred to beaded tubes (A29158, Thermofisher). Cells were lysed using BeadBug™6 homogenizer at 3500 rpm, 3 cycles 30 s each, suspensions were kept on ice between cycles for 1 min. Tubes were centrifuged at 9800× *g* for 15 min to remove cell debris and the supernatants were transferred to fresh microcentrifuge tubes. The level of catalase activity in cell lysates have been determined using Catalase assay kit (CAK1061, Cohesion Biosciences, UK). Total antioxidant capacity in Trolox equivalents have been measured using Total Antioxidant Capacity Assay Kit (MAK187, Sigma-Aldrich).

### 2.8. Maintenance of Cell Lines

Human Colorectal Adenocarcinoma Cell Line HT-29 and mucous-secreting HT-29-MTX (both purchased from Sigma-Aldrich) were propagated using low glucose DMEM medium supplemented with 10% Fetal Bovine Serum, 2 mM L-glutamine, 100 U/mL penicillin, and 100 µg/mL streptomycin in a 5% CO_2_ atmosphere at 37 °C. All cell culture reagents were purchased from Capricorn Scientific, Ebsdorfergrund, Germany. Pro-monocytic human cell line U937 obtained from the American Type Culture Collection (ATCC, CRL-1593.2) and was routinely cultured in RPMI-1640 medium containing 10% FBS, 2 mM L-glutamine, 100 U/mL penicillin, and 100 µg/mL streptomycin under conditions indicated above. All base media and supplements were purchased from Thermofisher.

### 2.9. Cell Viability Assays

Cell viability assays were carried out using CyQUANT™ XTT Cell Viability kit (Invitrogen™), according to manufacturer’s instructions. Prior to experiments, HT-29 cells were seeded onto 96-well plate at a density 1 × 10^5^ cell/well. The day after seeding, cells were washed twice with 200 µL of DPBS; 100 µL of prewashed bacteria (10^7^ CFU/mL–10^8^ CFU/mL) were added to the wells. Full culture medium containing 2.5% ethanol was added to the control wells (positive control). Twenty hours after exposure to bacteria HT-29 cells were washed twice with 200 µL of DPBS; 100 µL of full medium containing antibiotics were added to the wells. HT-29 cells were then stained with 0.3 mg/mL solution of XTT (sodium 3′-[1-(phenylaminocarbonyl)-3,4-tetrazolium]-bis (4-methoxy6-nitro) benzene sulfonic acid hydrate) for 4 h. Absorbance was detected at 450 and 595 nm.

In a parallel experiment, HT-29 cells pretreated with *B. clausii* CSI08 were subjected to three rounds of washing with 200 µL of DPBS; after that HT-29 cells were exposed to H_2_O_2_ added to a final concentration of 4 mM. After 24 h of incubation staining with XTT solution was performed as described above.

### 2.10. Adhesion Assays

HT-29-MTX cells were seeded onto 24-well plates at a density of 5 × 10^5^ cell/well and cultured for 21–28 days to complete maturation. Media was replaced every 2–3 days. Four hundred microliters of full media (low glucose DMEM, 10% Fetal Bovine Serum, 2 mM glutamine) without antibiotics were added to the wells allocated for bacteria; DPBS was aspirated from the wells allocated for spores after the second round of washing. One hundred microliters of pre-washed bacteria in DMEM (2.0 × 10^7^ CFU/mL–1.2 × 10^8^ CFU/mL) or 500 µL of spores suspensions (4.0 × 10^7^–9.0 × 10^7^ CFU/mL) were added to the cells, mixed by a gentle swirl, and incubated for 2.5 h at 37 °C in a 5% CO_2_ atmosphere. Control wells not containing mammalian cells were prepared and incubated in parallel in the same way. Upon incubation HT-29-MTX cells were washed 4 times with 0.5 mL PBS. After that, 50 µL of Trypsin/EDTA solution (CC-5012, Lonza, Switzerland) and 50 µL of PBS were added to the wells and incubated for 10 min with gentle shaking (~100 rpm) at 37 °C. Fifty microliters of Trypsin/EDTA solution were added to control wells. Consequently, 450 µL of PBS were added to the wells with bacteria or spores, contents of the wells were transferred into microcentrifuge tubes with scrapping and subjected to three rounds of vigorous shaking 30 s each. Contents of control wells were transferred into microcentrifuge tubes and subjected to one round of shaking. Serial dilutions (plus dilutions of control wells) were prepared in PBS and plated onto PetriFilm™ or MRS agar plates for quantification of *B. clausii* CSI08 or *L. fermentum*.

### 2.11. Anti-Inflammatory Activity Assays

HT-29 cells were seeded onto 24-well plates at a density of 5 × 10^5^ cell/well. The day after seeding cells were washed twice with 0.5 mL DPBS; 0.4 mL of full cell culture media without antibiotics were added to the wells allotted for pretreatment with viable bacteria. Alternatively, 0.25 mL of full cell culture media containing antibiotics were added to the wells allotted for pretreatment with bacterial supernatants. One hundred microliters of pre-washed bacterial cells (10^8^ CFU/mL) or 0.25 mL of cell free supernatants were added to the corresponding wells. After 20 h of incubation (CO_2_ atmosphere at 37 °C), HT-29 cells were washed twice with 0.5 mL of DPBS and 0.5 mL of full cell culture medium containing antibiotics were added to the wells. Lipopolysaccharides from *E. coli* O111:B4 (LPS, Sigma L4391) were added to HT-29 cells pretreated with viable bacteria and cell free supernatants to final concentration of 15 ng/mL. Poly I:C (P9582, Sigma) was added to pretreated HT-29 cells to final concentration of 10 µg/mL. Three or four hours after adding LPS or poly I:C, cell culture supernatants were removed and HT-29 cells were lysed in the wells by adding 300 µL of lysis buffer supplied with Monarch Total RNA Miniprep Kit (NEB, MA, USA).

### 2.12. RNA Extraction, Reverse Transcription, and Quantitative Real-Time PCR

Total RNA was extracted from cell lysates using Monarch Total RNA Miniprep Kit (NEB, MA, USA), according to manufacturer’s instructions. Then, fifty microliters of nuclease-free water were taken to elute RNA. A Qubit™ RNA broad range kit was used to quantify RNA after isolation. Next, one microgram of total RNA was taken to set up reverse transcription reactions using Luna script RT Supermix kit (NEB, MA, USA). Real-time PCR reactions were set up using Luna^®^ Universal qPCR Master Mix (NEB, MA, USA) and appropriate primer pairs (see Appendix A) at a concentration of 200 nM using 1 µL of generated cDNA per 9 µL of master mix. The reactions were performed in duplicates using the following program: initial denaturation 95 °C 5 min, denaturation 94 °C 20 s, annealing 60 °C 20 s, extension 72 °C 20 s (40 cycles). The specificity of reaction products was confirmed by melting temperature analysis (from 70 °C to 95 °C in 0.5 °C/15 s increments). Quantification of target transcripts was done using *gapdh* as a normalizing house-keeping gene.

### 2.13. Human Nulcear Factor-κB (NF-κB) p65 Transcription Factor Activity Assay

HT-29 cells were seeded onto 6-well plates at a density of 2 × 10^6^ cell/well. The day after seeding, cells were washed twice with 2 mL of DPBS; 1.6 mL of full cell culture media without antibiotics and 0.4 mL of pre-washed as described above (see Section 2.1) *B. clausii* CSI08 suspension in DMEM were added to the corresponding wells. Two milliliters of media were added to the control wells. After 20 h of incubation (37 °C CO_2_ atmosphere), HT-29 cells were washed two times with 2 mL of DPBS and 2 mL of full media containing antibiotics were added to all the wells. LPS were added to the wells pretreated with bacteria and to the control wells to a final concentration of 15 ng/mL. HT-29 cells were collected with 1 mL of ice-cold PBS by scrapping 45 min after adding LPS. Nuclear fractions were extracted using ProteinExt^®^ Mammalian Nuclear and Cytoplasmic Protein Extraction Kit (TransGen Biotech, Beijing, China) according to the manufacturer’s instructions with one modification. Nuclear fractions were collected with 150 µL of NPEB buffer, instead of 500 µL indicated in the supplied manual. Total protein concentration was thereafter determined using Bicinchonic Acid Protein Assay Kit (B9643, Sigma). The nuclear fractions normalized by total protein concentration (~25 µg of protein) were taken to evaluate NF-κB activity using the NF-κB activity assay kit (TFEH-p65-1, RayBiotech, GA, USA) according to the manufacturer’s instructions. Experiments were performed two times with three or four technical replicates per assay.

### 2.14. Macrophage Differentiation and Challenge Study

U937 cells were seeded at a concentration of 10^5^ cells/well in 96-well plates and subjected to macrophage differentiation upon 50 ng/mL phorbol 12-myristate 13-acetate (PMA) treatment for 72–96 h. Macrophages were then stimulated with 10^8^ CFU/mL of *B. clausii* CSI08 or 5 ng/mL LPS in complete RPMI-1640 medium without antibiotics for 5 h at 37 °C. Afterwards, cell supernatants were harvested and stored at −20 °C until cytokine determination. The analysis was carried out using Luminex 200™ System according to the manufacturer’s instructions with a cytokine panel from Thermofisher including IL-1β, IL-18, IL-6, TNF-α, GM-CSF, G-CSF, IL-10, IL-1RA and EGF.

### 2.15. C. elegans Culture Conditions and Experiments

*Caenorhabditis elegans* strains N2, Bristol (wild type) was obtained from the *Caenorhabditis* Genetics Center at the University of Minnesota and maintained at 20 °C on Nematode Growth Medium (NGM) plates with *E. coli* strain OP50 as normal diet for nematodes. Overnight, *B. clausii* CSI08 culture was grown, as described in Section 2.1, and then centrifuged for 10 min at 4000× *g*. Supernatants were then removed, and bacterial pellets were washed once with saline solution. *C. elegans* wild-type strain (N2) cultured in NGM (control fed condition), or NGM supplemented with *B. clausii* CSI08 at two different doses (10^8^ and 10^9^ cells/plate). Vitamin C (10 µg/mL) was used as positive control. Worms were incubated in these conditions and after several days were submitted to an acute oxidative stress (2 mM H_2_O_2_) according to a previously published protocol [46]. Afterwards, viability of nematodes was determined in each fed condition. Experiments were performed in duplicate. Each experiment was conducted on 5 different plates, each containing 10 worms (50 worms/assay). The antioxidant activity (worm survival) of total population was calculated. Final survival data correspond to the average of two independent assays (total population of 100 worms/condition). The effect of *B. clausii* CSI08 antioxidant activity was studied by comparing the survival of treated nematodes versus the control-fed nematodes.

### 2.16. Statistical Analysis

All data were analyzed using Prism 9 (GraphPad Software, San Diego, CA, USA). Normal distribution was determined using Shapiro-Wilk test. Samples following normal distribution were tested for significance using unpaired *t* test or one-way ANOVA with Tukey, Bonferroni or Dunnets post-hoc as relevant. When samples did not follow normal distribution, a Mann-Whitney U test or Kruskal-Wallis with Dunn’s post-hoc was performed.

## 3. Results

### 3.1. In Vitro Evaluation of the Probiotic Properties of B. clausii CSI08

#### 3.1.1. Resistance to an In Vitro Simulated Gastric and Intestinal Conditions

The ability of *B. clausii CSI08* spores to survive during an in vitro simulated digestion process was compared to that of the commercial *L. rhamnosus* GG (ATCC 15103) strain. As shown in Figure 1, there was no decrease in the *B. clausii CSI08* spores count after exposure to simplified gastric and small intestinal conditions. At the same time, we detected a significant drop in viable bacterial counts at the intestinal stage (0.1% pancreatin, 0.3% bile salts). These data indicate the potential ability of *B. clausii CSI08* spores to efficiently survive the transit through the upper digestive tract.

#### 3.1.2. Stability of *B. clausii* CSI08 Spores during Pasteurization

The temperature stability of *B. clausii* CSI08 spores was assessed at various conditions (45, 75, and 90 °C) at three different time points (0.5, 1, and 3 min) in PBS. The summarized results are shown in Figure 2. No change in viability was detected at 45 °C or 75 °C at all three time points. Yet, we recorded the reduction in spore counts after treatment at 90 °C from the 6.87 × 10^9^ to 3.01 × 10^9^ CFU. Nevertheless, *B. clausii* CSI08 spores showed good overall performance at pasteurization conditions.

#### 3.1.3. Antimicrobial Activity

Antimicrobial activity *B. clausii* CSI08 in its vegetative form was evaluated for antimicrobial activity against four strains of *E. coli*, *S. enteritidis*, *S. aureus*, and *P. aeruginosa*. The results shown in Figure 3 demonstrate the reduction in bacterial counts of three tested pathogens grown in the presence of *B. clausii* CSI08. At the same time, under conditions indicated above *B. clausii*, CSI08 was unable to inhibit growth of *S. enteritidis*.

#### 3.1.4. Enzymatic Capacity of *B. clausii* CSI08

The carbohydrate assimilation pattern determined using the API 50 CH assay kit is shown in Appendix A. *B. clausii* CSI08 was able to metabolize various simple carbohydrates including glycerol, L-arabinose, D-ribose, D-glucose, D-fructose, D-mannose, L-rhamnose, as well as disaccharides, including saccharose, cellobiose, trehalose, with moderate ability to metabolize polymeric amidon and glycogen.

Enzymatic profile of *B. clausii* CSI08 detected using the API ZYM kit is presented in Appendix A. We detected activities of esterase-lipase C8 and esterase C4, hydrolyzing lipids into free glycerol and unsaturated and saturated fatty acids. Notably, *B. clausii* CSI08 was able to produce β-galactosidase, the enzyme that catalyzes lactose hydrolysis into glucose and galactose and, importantly, is responsible for formation of galacto-oligosaccharides (GOS) promoting the growth of *Bifidobacterium* and *Lactobacillus* species [47].

#### 3.1.5. Amino Acids and Water-Soluble Vitamins in *B. clausii* CSI08 Supernatants

Furthermore, we performed an exploratory study to assess the ability of *B. clausii* CSI08 to produce amino acids and vitamins. The levels of 22 amino acids and water-soluble vitamins were determined in overnight cultures of *B. clausii* CSI08. The analysis showed the higher levels of glutamic acid, alanine, glutamine and histidine mixture, threonine, proline, tyrosine, valine, and methionine in bacterial supernatants compared to TSB medium (Figure 4a). These data suggest the capacity of *B. clausii* CSI08 to synthesize the above-mentioned amino acids during their growth in rich medium. At the same time, the elevated concentration of amino acids might be the result of a peptidase activity of the strain. We further detected high levels of pantothenic acid (B5) and cobalamin (B12) in *B. clausii* CSI08 supernatants (Figure 4b), which supports the assertation of the strain to produce vitamins.

#### 3.1.6. Assessment of Safety of *B. clausii* CSI08 and Adhesion to Mucous-Producing Cell Line HT-29-MTX

To evaluate any potential cytotoxic effect of vegetative cells *B. clausii* CSI08 on HT-29 intestinal cells, the XTT assay has been carried out 20 h after bacterial exposure. Approximate multiplicity of infection (MOI) in the assay constituted 1:500. As shown in Figure 5a, *B. clausii* CSI08 did not negatively impact the survival of intestinal epithelial cells at given MOI. The adhesion ability of *B. clausii* CSI08 and its spore preparations were compared to that of a commercial probiotic *L. fermentum* strain (Figure 5b). Both forms (cells and spores) of *B. clausii* were able to adhere to HT-29-MTX cells. Percentage of adhesion of the vegetative cells was approximately three times higher than that of the spores. *L. fermentum* was shown to adhere to the mucous-producing cells twice as efficiently as vegetative cells of *B. clausii*.

### 3.2. Immunomodulatory Effect of B. clausii CSI08 in Human Cell Lines

The vegetative form of *B. clausii* CSI08 and its cell-free supernatants were assessed for their ability to modulate pro-inflammatory response in HT-29 cell line, triggered by *E. coli* lipopolysaccharides (LPS) or Poly I:C, a synthetic double-stranded RNA ligand. Expression level of IL-8, TNF-α, IL-17C, CXCL10 genes have been evaluated at mRNA level using qRT-PCR. As shown in Figure 6, expression of all 4 genes was strongly induced 4 h after stimulation with LPS. Pretreatment with *B. clausii* CSI08, but not with its cell-free supernatants, greatly inhibited LPS-triggered pro-inflammatory response. Notably, neither the vegetative cells of *B. clausii* or their supernatants by themselves elicited a significant increase in IL-8, TNF-α, IL-17C, CXCL10 gene expression after co-incubation with HT-29 cells (Figure 6).

When the pro-inflammatory changes in HT-29 cell line were triggered by Poly I:C, both cells and cell-free supernatants of *B. clausii* CSI08 have been shown to attenuate the overexpression of IL-8, TNF-α, IL-17C, and CXCL10 genes (Figure 7).

In order to confirm that inhibitory effect of *B. clausii* CSI08 on the expression of proinflammatory marker genes was associated with attenuation of NF-κB, we investigated the activity of the transcription factor in the nuclear fractions of HT-29 cells preincubated with viable bacteria 45 min after adding LPS. As shown in Figure 8, *B. clausii* CSI08 significantly counteracted the LPS-triggered activation of NF-κB.

Subsequently, we investigated the effect of *B. clausii* CSI08 on the innate immune system, adopting the cell model of U937-derived macrophages. The latter were stimulated with 10^8^ CFU/mL of vegetative cells *B. clausii* CSI08, followed by quantification of cytokines/chemokines 5 h after exposure to bacteria. The concentrations of pro-inflammatory TNF-α, IL-1β, IL-18, regulatory G-CSF, GM-CSF, IL-6, and anti-inflammatory IL-10, IL-1RA, EGF were determined. The response to *E. coli* lipopolysaccharides has been assessed in parallel experiments. As shown in Figure 9, *B. clausii* CSI08 provoked a robust cell response, resulting in secretion of high levels of all analyzed cytokines.

### 3.3. Antioxidant Capacity of B. clausii CSI08 In Vitro and In Vivo

The total antioxidant capacity and the catalase activity of *B. clausii* CSI08 cell lysates were compared to those of *L. rhamnosus* GG. Strong antioxidant properties of the abovementioned strain have been acknowledged by a number of studies [48]. *B. clausii* CSI08 and *L. rhamnosus* GG exhibited a similarly high level of total antioxidant activity (Figure 10a). In contrast, the level of catalase activity was significantly higher in *B. clausii* CSI08 cell lysates as compared with the reference *L. rhamnosus* strain (Figure 10b).

The antioxidant potential of *B. clausii* CSI08 has been confirmed using the cell model of oxidative damage. The viability rate of HT-29 epithelial cells treated with hydrogen peroxide has been compared with that of the cells preincubated with *B. clausii* CSI08 taken at three different concentrations prior to adding H_2_O_2_. A significant reduction in viability was observed in HT-29 cells exposed to H_2_O_2_. Pretreatment with 2.0 × 10^8^, 1.0 × 10^8^, 5.0 × 10^7^ CFU of *B. clausii* CSI08 partially restored the impaired viability of epithelial cells. Summarized results are shown in Figure 10c. To further characterize antioxidant properties of *B. clausii* CSI08, the in vivo *C. elegans* model was employed. Nematodes were fed with vegetative cells at two different concentrations and after that subjected to acute oxidative stress with hydrogen peroxide. The survival rate was compared to that of the control group nematodes (fed with nematode growth medium only). As shown in Figure 10d, *B.clausii* CSI08 added to the worm diet caused an increase in the survival of the *C. elegans* population after exposure to oxidative stress in a dose-dependent manner, being 10^9^ cells/ml the functional dose, with a survival of 44% compared to 29% of the control fed condition.

## 4. Discussion

Spore-forming bacteria of the genus *Bacillus* represent an ample and remarkably diverse group of microorganisms. Their capacity to produce a wide variety of secondary metabolites, such as antibiotics, enzymes, vitamins, and carotenoids, created a huge potential for use in pharmaceutical, agricultural, and industrial processes [49,50,51]. *Bacillus* strains have been recognized as safe and efficacious probiotics, offering, in addition to classic probiotic traits, notably high stability during processing and storage [52], as well as relatively low production costs [53]. Microorganisms of *B. clausii* group stand out from other *Bacillus* species featuring a distinctive ability to modify immune response in vitro and in vivo [54]. In the present work we introduced the novel strain of *B. clausii* CSI08, with a focus on immunomodulatory properties of this potential probiotic.

The ability to withstand low pH and the presence of high concentration of bile salts is one of the fundamental properties of probiotic strains [55]. Generally recognized as being highly resistant to extreme environmental stress, spores can struggle to maintain viability after exposure to bile salts [56]. Therefore, assessment of spores’ survival under digestion-related stress is a crucial step in a spore-based probiotic selection pipeline. *B. clausii* CSI08 spores demonstrated excellent resistance to the simulated gastric and small intestinal environment, indicating their potential ability to survive passage through upper gastrointestinal tract. Previous findings have demonstrated that spore formulations of the well-known *B. clausii* probiotic strains can tolerate pH 2 and up to 1% conjugated bile salts and, moreover, can germinate and multiplicate under mimicked human intestinal environment [39,57,58]. Similarly, we demonstrate herein that *B. clausii* CSI08 spores can survive simulated gastric and small intestinal conditions.

Adherence to the intestinal mucosa is another essential characteristic of potential probiotics. Stable binding to the intestinal epithelium helps bacteria and spores to avoid their quick removal by peristalsis. Even transient colonization of mucosa by probiotic strains leads to competitive exclusion of pathogens [59] and allows them to interact with the local immune system [60]. Given the critical role of mucins in adhesion [61,62], we have investigated the ability of vegetative cells and spores of *B. clausii* CSI08 to adhere to the mucous-producing cell line HT-29-MTX. *B. clausii* CSI08 in both live states showed the strong adhesive capacity, somewhat lower than that of the control *L. fermentum* strain. In the study published by Ahire JJ et al. [58], spores of *B. clausii* UBBC07 demonstrated significantly higher adhesion levels to porcine mucin compared to vegetative cells, conversely to our results. Genome sequencing analysis of the probiotic *B. clausii* strains [37,38] revealed the presence of genes encoding for adhesion-related proteins such as mucus-binding protein [63], enolase [64], flagellar hook-associated protein [65], and exopolysaccharide biosynthetic gene clusters [66]. Moreover, strains of *B. clausii*/ Enterogermina™ have been recovered from feces of healthy volunteers twelve days after single administration of probiotic preparations, which indirectly suggests the ability of *B. clausii* strains to temporarily colonize the intestine [67].

The enzymatic profiling conducted In the course of present study revealed β-galactosidase activity of *B. clausii* CSI08. Release of this enzyme, responsible for the hydrolysis of lactose, is an important feature of probiotics, since it has been involved in the relief of lactose intolerance symptoms [68]. Another significant trait of *B. clausii* CSI08 is the lack of detrimental β-glucuronidase, α-chymotrypsin, and β-glucosaminidase activities. Beta-glucuronidase plays a critical role in colorectal carcinogenesis, being implicated in reactivation of carcinogen metabolites [69,70], α-chymotrypsin and β-glucosaminidase, produced by gut bacteria, are among important factors in the pathogenesis of endocarditis [71].

Antimicrobial activity is another undoubtedly essential feature of potentially probiotic bacterial strains. We showed the capacity of *B. clausii* CSI08 to inhibit the growth of opportunistic pathogens, *E. coli*, *S. aureus*, and *P. aeruginosa*, during co-cultivation in liquid culture. Probiotic strains of *B. clausii* are known to produce substances with antimicrobial activity; some of them have been identified and characterized. Biological target of the lantibiotic clausin, first discovered in the cell-free supernatants of *B. clausii* O/C, are lipid intermediates essential for the biosynthesis of peptidoglycan and other bacterial cell wall polymers [43,72]. Clausin has been shown to be active against a range of Gram-positive bacteria including *C. difficile* [39,43]. Recently, Ahire JJ et al. demonstrated the production of class I lantibiotic clausin by *B. clausii* UBBC07 in the in vitro Simulator of Human Intestinal Microbial Ecosystem (SHIME) model [39]. Interestingly, production of antimicrobial substances active against Gram-negative and Gram-positive species have been observed upon whey fermentation by *B. clausii* [36], indicating the importance of starting substrates for synthesis of antimicrobials. Another remarkable example of indirect antimicrobial activity has been reported by Ripert G et al. in 2016 [35]. The study describes the serine protease produced during the sporulation phase by *B. clausii* O/C that can abolish the cytotoxic effects of toxigenic strains of *C. difficile* and *B. cereus*.

It has been shown that gut bacteria can influence the bioavailability of amino acids to the host organism [73]. Adsorption of amino acid is highly efficient in the small intestine [74], whereas in the colon amino acids are effectively metabolized by the microbiota [75]. The metabolic end products amongst others include short chain fatty acids and branched-chain fatty acids, that in turn significantly effect physiology of epithelial cells and the mucosal immune system [76]. We have demonstrated that growth of *B. clausii* CSI08 in rich medium was associated with an increase in concentration of eight essential amino acids. Further studies are needed to confirm the nature of the observed effect (de-novo synthesis or proteolytic activity). Furthermore, we have reported that *B. clausii* CSI08 can potentially synthesize pantothenic acid and cobalamin. Recent studies suggest an immunostimulatory role of pantothenic acid in the context of anticancer therapy [77]. Interestingly, vitamin B5 and its metabolite CoA, vital for metabolism of fatty acids, have been recently shown to increase mitochondrial metabolism of T cells and subsequently their anti-tumor efficacy [78]. Vitamin B12, amongst multiple biological functions, have been shown to regulate cellular immunity by increasing activity of CD8+ T lymphocytes and natural killer (NK) cells [79].

Disruption of the normal microbial community structure, environmental factors such as stress, and diet, as well as genetic factors, can trigger chronic intestinal inflammation [60]. The ability of *B. clausii* CSI08 to attenuate inflammatory response induced by LPS and Poly I:C, interacting with TLR4 [80] and TLR3 [81] receptor complexes correspondingly, was evaluated in the course of the present work. Pretreatment of HT-29 cells with vegetative *B. clausii* CSI08 attenuated the LPS-induced expression of pro-inflammatory (IL-8, TNF-α, IL-17C) genes. We also confirmed that this effect was associated with down-regulation of the canonical NF-κB transcription factor [82]. Both cells and cell-free supernatants of *B. clausii* CSI08 attenuated the pro-inflammatory response triggered by Poly I:C. Similarly, the protective effect of *B. clausii* (O/C, T, SIN and N/R) strains against Rotavirus-induced changes in Caco-2 cell line of human enterocytes have been demonstrated in the recent study published by Paparo L et al. [83]. Moreover, in a murine model of chronic colitis administration of *B. clausii* (O/C, T, SIN and N/R) resulted in reduction of the colonic inflammatory score [84]. Another strain of *B. clausii,* MTCC8326, was found to be effective in attenuating *Salmonella typhimurium*-associated dysbiosis and inflammation in Th2 (BALB/c)-biased mice [85].

It has been proven that certain probiotic strains can enhance innate immune response via stimulation of macrophages and dendritic cells located in the lamina propria [86,87,88]. Vegetative cells of *B. clausii* CSI08 were able to elicit the robust response in U937-derived macrophages, resulted in markedly increased production of pro-inflammatory (TNF-α, IL-1β, IL-18) and anti-inflammatory (IL-10, IL-1RA, EGF) cytokines. This is in agreement with Pradhan B et al. [41] who observed the induction of a pro-inflammatory response at earlier time points and an anti-inflammatory response at later time points after exposure murine RAW 264.7 macrophages to *B. clausii* MTCC-8326. Villéger et al. [89] have proposed the essential role of D-alanine of lipoteichoic acids for the immunomodulatory properties of probiotic strains of *B. clausii.* Another remarkable example of involvement of *B. clausii* in the immune activation, that can influence clinical outcomes in patients with pancreatic adenocarcinoma, has been described by Riquelme E et al. [90].

The antioxidant capacity of probiotics and potential health-promoting effects related to it have been widely studied in recent decades [91]. *B. clausii* CSI08 showed strong antioxidant activity in vitro, and, moreover, in its vegetative form exerted cytoprotective effect in cell culture and *C. elegans* models of oxidative stress, likely through the reduction of reactive oxygen species. Publications demonstrating the antioxidant capacity of *B. clausii* strains are limited and reported only in cell lines and murine studies. In a rat model of uremia, administration of *B. clausii* UBBC07 was associated with alleviation of oxidative response induced by acetaminophen, evidenced by increased levels of glutathione, superoxide dismutase, and catalase [92]. Similarly, in a rat trinitrobenzenesulfonic acid (TNBS)-induced colitis model treatment with *B. clausii* was partially protective against oxidative damage [93]. Finally, *B. clausii* have been shown to suppress Rotavirus-induced production of reactive oxygen species in Caco-2 cell line [83].

## 5. Conclusions

The study presented the novel strain of *B. clausii* CSI08 with prominent probiotic characteristics. Spore preparations of *B. clausii* CSI08/Munispore^®^ demonstrated high level of resistance to gastric pH and high bile salts concentration. The vegetative form of the strain showed significant ability to suppress the growth of pathogenic bacteria in liquid culture, was capable of assimilating a wide range of carbohydrates and was shown to produce potentially beneficial enzymes. *B. clausii* CSI08 did not demonstrate cytotoxic effect towards intestinal epithelial cells and displayed moderate adhesion capacity in both live states. Additionally, we showed that spores of *B. clausii* CSI08 can effectively survive an in vitro simulated pasteurization process. Furthermore, *B. clausii* CSI08 exhibited a strong ability to attenuate LPS- and Poly I:C-triggered inflammatory response in the intestinal epithelial cell line and to enhance the innate immunity via macrophage stimulation. Moreover, this novel strain displayed excellent antioxidant characteristics when studied in vitro and in vivo. Taken together, the data herein provide strong evidence for potential therapeutic efficacy for *B. clausii* CSI08 as a promising probiotic strain.

## Figures and Tables

**Figure 1 microorganisms-11-00240-f001:**
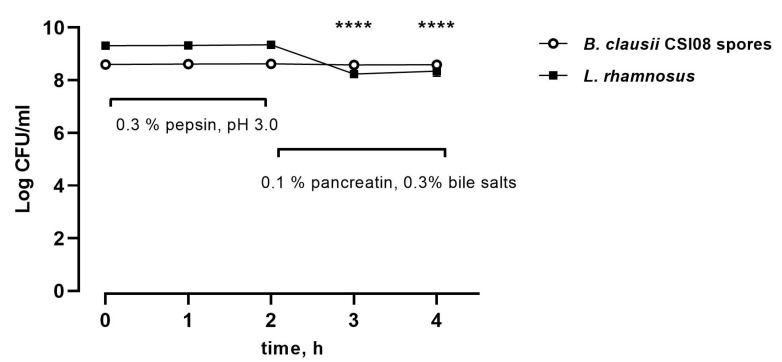
Survival of *B. clausii* CSI08 spores and *L. rhamnosus* ATCC 53103 at gastric and small intestinal digestion conditions simulated in vitro. The values are expressed in Log10 CFU/mL. Data represent the mean (n = 8) ± SEM of two independent experiments performed in four technical replicates. **** *p* < 0.0001 Survival of *L. rhamnosus* compared to 0 h.

**Figure 2 microorganisms-11-00240-f002:**
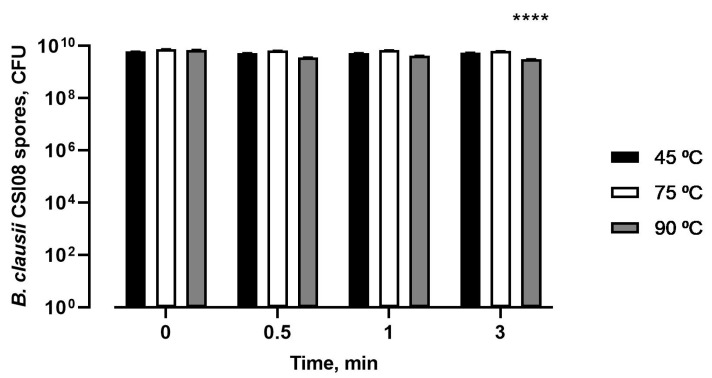
Thermostability of *B. clausii* CSI08 spores at 45, 75, and 90 °C at different time intervals. The values are expressed in CFU of spores. Data represent the mean (n = 9) ± SEM of three independent experiments performed in triplicate. Spore counts compared to 0 min. **** *p* < 0.0001.

**Figure 3 microorganisms-11-00240-f003:**
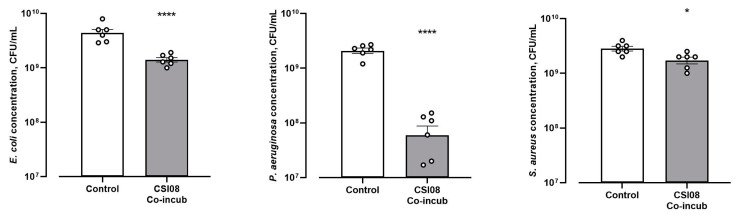
Antimicrobial activity *B. clausii* CSI08 in liquid culture. Values represent average concentration ± SEM of two independent experiments with three technical replicates (n = 6). * *p* < 0.05, **** *p* < 0.0001.

**Figure 4 microorganisms-11-00240-f004:**
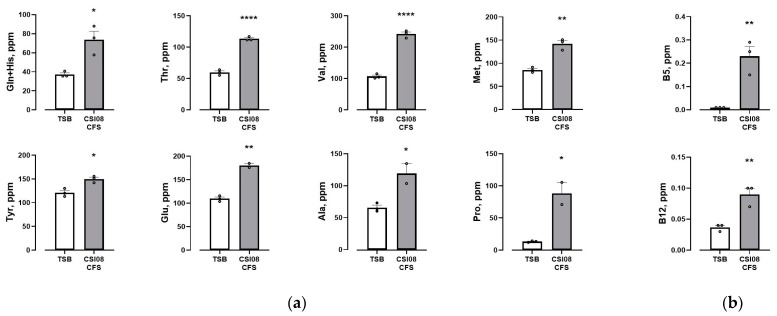
(**a**) Amino acids and (**b**) vitamins in *B. clausii* CSI08 supernatants (CFS) determined by HPLC-FLD and UHPLC-MS correspondingly. Values are means (n = 3) ± SEM. * *p* < 0.05, ** *p* < 0.01, **** *p* < 0.0001.

**Figure 5 microorganisms-11-00240-f005:**
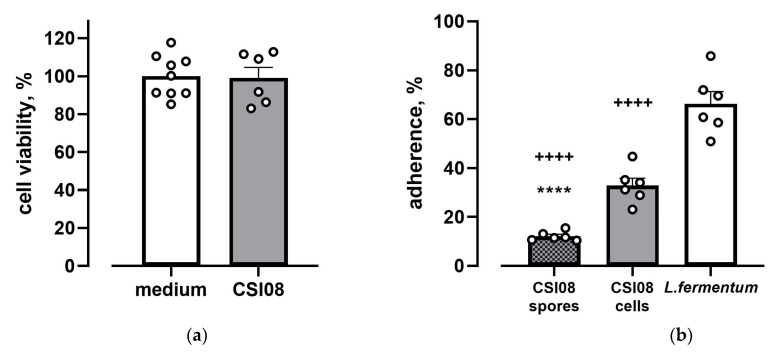
Interaction of *B. clausii* CSI08 with intestinal epithelial cells. (**a**) *B. clausii* CSI08 did not reduce HT-29 viability. Data represent percentage of XTT dye conversion after 20 h of coincubation with *B. clausii* CSI08 compared with untreated cells (medium). Values are means ± SEM of three independent experiments. (**b**) Adhesion of vegetative cells and spores *B. clausii* CSI08 to the HT-29-MTX. Values represent percentage of adhered cells/spores after 2.5 h of incubation with HT-29-MTX followed by four rounds of washing. Values are means (n = 6) ± SEM of three independent experiments **** *p* < 0.0001 vs. CSI08 cells, ++++ *p* < 0.0001 vs. *L. fermentum*.

**Figure 6 microorganisms-11-00240-f006:**
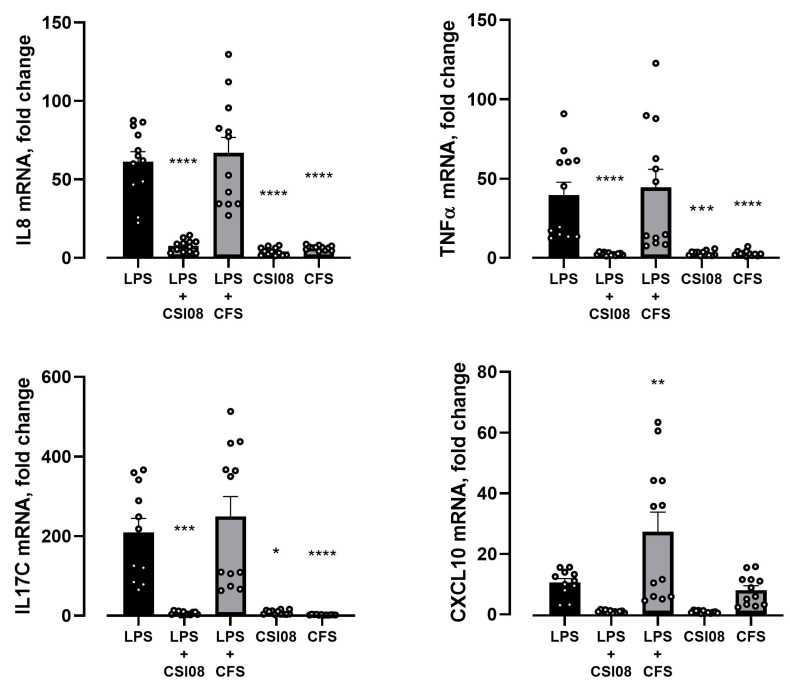
*B. clausii* CSI08 attenuated LPS-induced pro-inflammatory response in HT-29 cell line. qPCR analysis of IL-8, TNF-α, IL-17C, and CXCL10 gene expression 4 h after exposure to 15 ng/mL LPS in HT-29 cells preincubated with *B. clausii* CSI08 or its cell free supernatants (CFS). * *p* < 0.05 vs. LPS, ** *p* < 0.01 vs. LPS, *** *p* < 0.001 vs. LPS, **** *p* < 0.0001 vs. LPS. The pattern of gene expression determined after co-incubation of HT-29 cells with *B. clausii* CSI08 and its CFS in unstimulated conditions is also shown. Results show mean ± SEM of three independent experiments.

**Figure 7 microorganisms-11-00240-f007:**
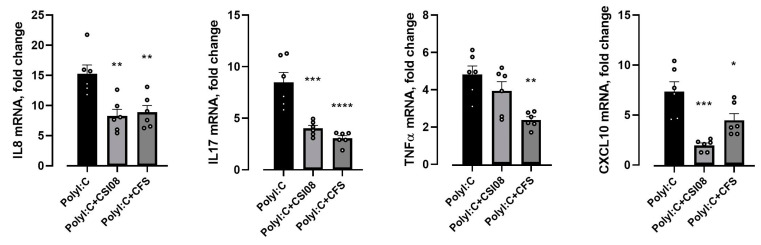
*B. clausii* CSI08 attenuated the Poly I:C-triggered pro-inflammatory response in HT-29 cell line. qPCR analysis of IL-8, TNF-α, IL-17C, and CXCL10 gene expression 4 h after exposure to 10 µg/mL Poly I:C in HT-29 cells preincubated with *B. clausii* CSI08 or its cell free supernatants (CFS). * *p* < 0.05, ** *p* < 0.01; *** *p* < 0.001, **** *p* < 0.0001. Results show mean (n = 6) ± SEM.

**Figure 8 microorganisms-11-00240-f008:**
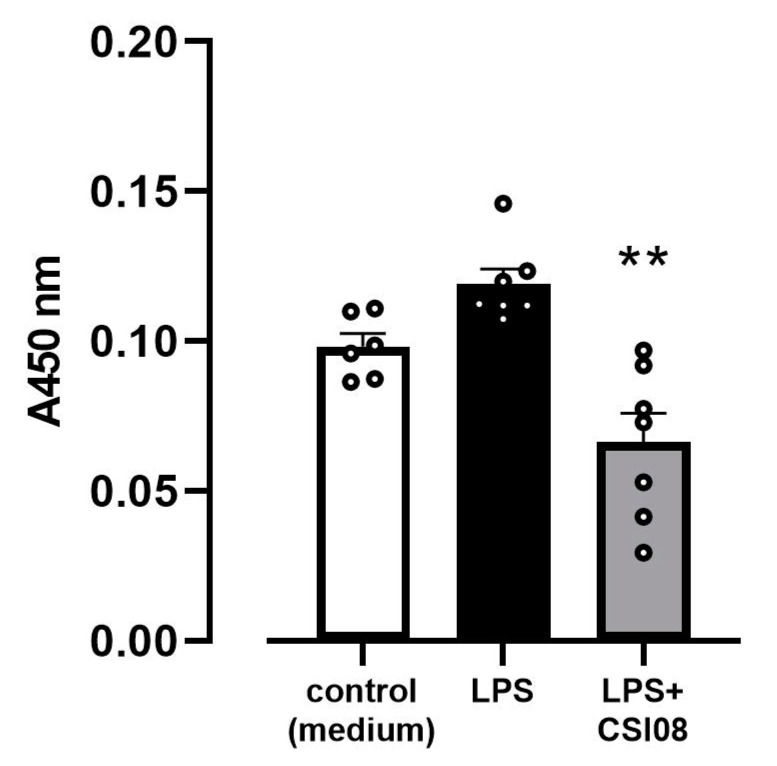
*B. clausii* CSI08 down-regulated the LPS-stimulated activation of NF-κB in HT-29 cell line. Adsorption values correspond to the levels of the transcription factor in the nuclear fractions of control cells, cells exposed to 15 ng/mL of LPS for 45 min, cells pretreated with *B. clausii* CSI08 for 20 h prior to adding LPS. Values are means ± SEM of two independent experiments. ** *p* < 0.01 vs. LPS.

**Figure 9 microorganisms-11-00240-f009:**
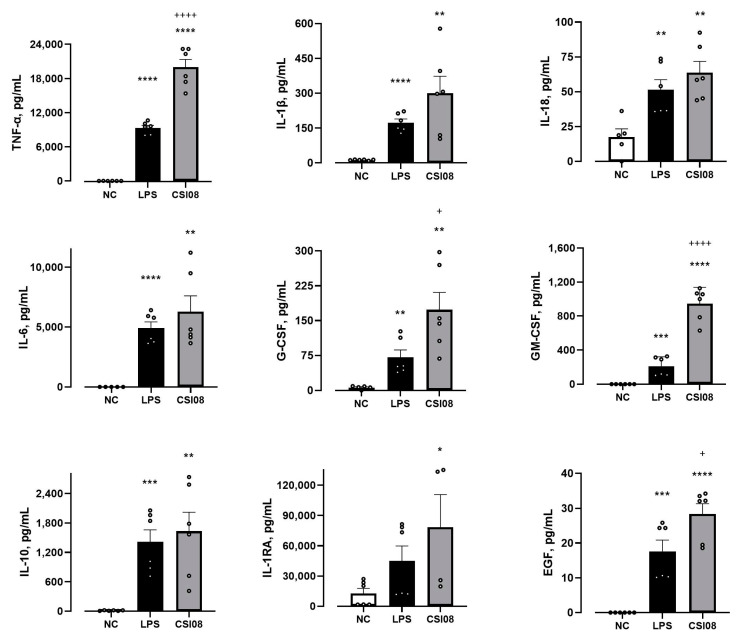
Immunostimulatory effect of *B. clausii* CSI08. Cytokine levels in cell culture supernatants of U937-derived macrophages: untreated (NC) or challenged with 10^8^ CFU/mL of vegetative cells of *B. clausii* CSI08 or LPS for 5 h. Values are the means ± SEM of three independent experiments. * *p* < 0.05, ** *p* < 0.01, *** *p* < 0.001; **** *p* < 0.0001 LPS or CSI08 vs. NC; + *p* < 0.05; ++++ *p* < 0.0001 CSI08 vs. LPS.

**Figure 10 microorganisms-11-00240-f010:**
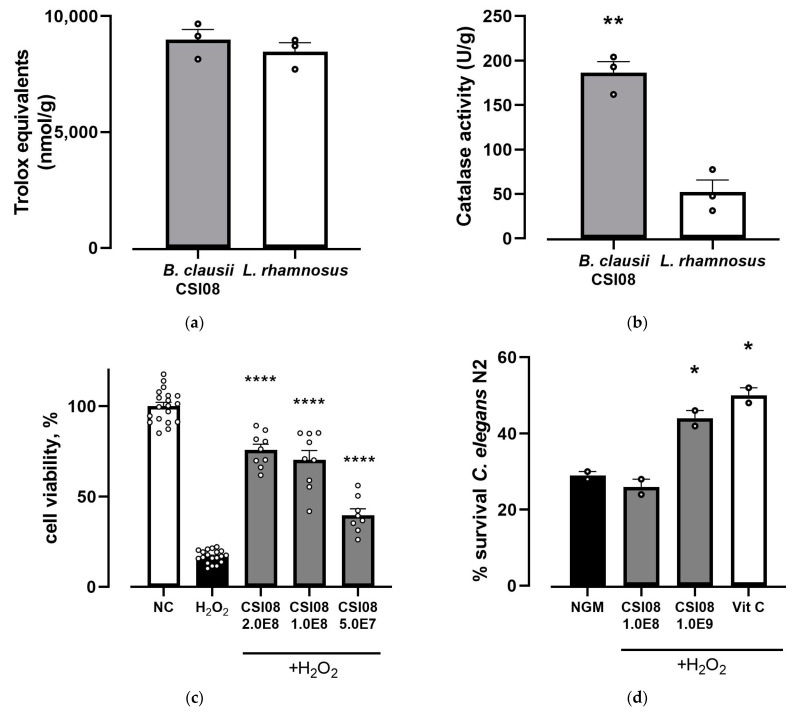
(**a**) Total antioxidant capacity of *B. clausii* CSI08 and *L. rhamnosus* GG cell lysates in Trolox equivalents (nmol/g). Presented values are means (n = 3) ± SEM. (**b**) *B. clausii* CSI08 demonstrated superior catalase activity (in U/g) as compared with *L. rhamnosus* GG cell lysates. Values are means (n = 3) ± SEM. ** *p* < 0.01. (**c**) Cytoprotective effect of *B. clausii* CSI08 on H_2_O_2_-exposed epithelium. Data represent percentage of XTT dye conversion 20 h after exposure to 4 mM H_2_O_2_ by HT-29 cells preincubated with 2.0 × 10^8^, 1.0 × 10^8^, 5.0 × 10^7^ CFU of *B. clausii* CSI08 compared with control group (NC). The values are the means ± SEM. **** *p* < 0.0001 vs. H_2_O_2_. (**d**) Survival rate of *C. elegans* N2 fed with 10^8^ and 10^9^ CFU/mL *B. clausii* CSI08 followed by an acute oxidative stress caused by 2 mM H_2_O_2_. NGM—control-fed nematodes; Vitamin C as a positive control. Values presented are the average of two independent experiments (n = 100/condition). * *p* < 0.05.

## Data Availability

Results of all analyses are included in this published article. The datasets generated and/or analyzed during the current study are available from the corresponding author on reasonable request.

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
