# Peer review of "Immunomodulatory and Antioxidant Properties of a Novel Potential Probiotic Bacillus clausii CSI08"

_microorganisms, 2023, doi:10.3390/microorganisms11020240_

Round 1
Reviewer 1 Report
The authors presented beneficial properties provide strong evidence for B. clausii CSI08 as a promising potential probiotic. The introduction, methodology, results, and conclusion were clear and appropriate for the journal. I would like to ask the authors to add a futuristic objective for this bacterium in animal models and how they compare the potential in studying the in Vivo effects of the probiotic stain.
Reviewer 2 Report
This manuscript verified the stability of probiotic strain Bacillus clausii CSI08 (the ability to survive in the upper digestive tract during transport and the stability under pasteurization conditions), verified its ability for anti-microbial activity against three strains of E. coli, S. aureus, and P. aeruginosa and the ability to produce vitamins. They also verified that this novel strain displayed excellent antioxidant characteristics when studied in vitro and in vivo. The whole subject is meaningful and worth of study. I feel that it is suitable for publication in this journal, however, the authors should accept some revisions of their paper, particularly on the following points:
1. L96, page 2: Munispore® containing probiotic strain Bacillus clausii CSI08. It is a compound probiotic. What is the proportion of B. clausii CSI08? Did they successfully screen out B. clausii CSI08 strains? Whether the experiment can prove that B. clausii CSI08 can play a major role or a supporting role in promoting the growth of other beneficial bacteria.
2. L118, page 3 temperature notation. NOT angle symbol.
3. L224, page 5: “poly I·C” to “poly I:C”
4. L310, page 7: “0.5-, 1-, 3 min durations”, please check with L118 page 3: “0.5, 1 or 3-minutes”.
5. L311, page 7: temperature notation
6. L348, 352 and 354, page 9: Will symbol unification be better? For instance, symbols indicating significant differences (p value) may be unified in degree by *, * *, * * *, * * * *, indicating from small to large differences, or vice versa.
7. L348 and 352, page 9: Please note that the letter size (Figure A B C or a b c) should be consistent.
8. L368, page 9: Why is the evaluation of cell adhesion ability of B. clausii CSI08 compared with Lactobacillus fermentum? Is the adhesion ability of B. clausii CSI08 lower than that of Lactobacillus fermentum good or bad ?
9. It is suggested to supplement some in vivo experiments. Is simulated in vivo experiments sufficient to prove the effectiveness of Bacillus clausii CSI08 strains / spores? Have the in vivo experiments in the cited literature been validated?
10. It is suggested to consider other real complex human environmental factors. No data in the real human intestinal environment was recorded in all experiments. Will B. clausii CSI08 strains / spores react with microbes colonized in gastrointestinal tract?
Round 2
Reviewer 2 Report
I feel that it is suitable for publication in this journal. Thank you for your work and answers.